# Two-Dimensional Attenuation and Velocity Tomography of Iran

**Thomas M. Hearn** 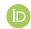

Physics Department, New Mexico State University, Las Cruces, NM 88003, USA; thearn@nmsu.edu

**Abstract:** Seismic bulletin data collected by the Iranian Seismological Center are used to image crust and mantle seismic attenuation, group velocity, and phase velocities for Lg, Pg, Sn, and Pn phases. This is possible because the peak amplitude time is picked, and amplitude measurements can be associated with the phase based on travel time plots. The group velocity is the apparent velocity of the maximum amplitude arrival and represents the combined effect of phase velocity and seismic scattering. Thus, it can be used in combination with the attenuation to identify where scattering attenuation is dominant. The Arabian–Iranian plate boundary separates low-velocity Zagros sediments from central Iran; however, in the mantle, it separates a high-velocity Arabian shield from central Iran. Scattering attenuation is low within the Arabian mantle and crust, and the Zagros sediments do not cause Lg or Pg attenuation. The Eocene Urumieh Dokhtar Magmatic Arc has high attenuation within both the crust and mantle, and while there is no partial melting in the crust, there may be some in the mantle. The northern Eocene Sistan Suture Zone shows particularly high attenuation that is accompanied by high scattering. It represents an incompletely closed ocean basin that has undergone intense alteration. The Alborz Mountains have high attenuation with some scattering.

**Keywords:** tomography; attenuation; Iran

## 1. Introduction

Understanding Earth's crustal structure requires the investigation of both attenuation and seismic velocity using a variety of phases. The attenuation of regional seismic amplitudes is important for understanding Earth structure, estimating seismic magnitudes, hazard analysis, and for nuclear seismic monitoring purposes. It is sensitive to temperature, partial melt, and rock composition and heterogeneity. It is known that volcanic regions and sedimentary basins can attenuate regional seismic waves [1–3]. Importantly, attenuation information complements those found in velocity studies. Low velocity combined with high attenuation can indicate partial melt, and high velocity combined with low attenuation indicates older cratonic terrains. In this paper, attenuation and velocity tomography are applied to investigate the structure of Iran.

The structure of the Iranian Plateau has been of much interest in seismology due to its rich tectonic history (Figure 1). It represents part of the closure of the Paleotethys and Neotethys oceans and the subsequent collision of the Arabian and Iranian plates. The Alborz mountains in the north first formed during the Triassic because of a collision between the Turan Plate and the Iranian Plate as the Paleotethys closed; later, deformation and volcanism were due to the Oligocene collision of the Arabian Plate with the Iranian plate as the Neotethys closed. The convergence there is around 10 mm/yr [4]. The Zagros Mountains in the south also formed during this later collision. They consist of over 10 km of folded passive ocean margin sediments [5]. The northeast edge of the Zagros, the Main Zagros Fault, marks the plate boundary. Current convergence rates across the Zagros Mountains are about 10 mm/yr [4]. The oceanic slab that subducted beneath the Zagros fell off about 10–15 My ago [6–9], causing plateau uplifts and spawning Neogene volcanism. There are many volcanoes in Iran, but the most recent are Damavand, Sabalan, and Sahand,

all in the Alborz, and the Qal'eh Hasan Ali maars. Sabalan shows some calc-alkalic volcanism, while the others have alkali volcanism, but it is not clear how they are related to past subduction [10]. While these show geothermal activity and Holocene volcanism, there is no historical volcanism. The Eocene was a time of tremendous volcanism in Iran (Figure 1) and amongst the features created is the Urumieh Dokhtar Magmatic Arc, which was an arc that formed due to the subduction of the Arabian Plate prior to continental collisions [5,11]. Eocene volcanism also occurred in the eastern Iranian ranges, which are also known as the Sistan Suture. It formed when a branch of the Neotethys ocean called the Sistan Ocean closed [12]. They are bordered by north–south right-lateral faults [4]. The average Moho depths across much of Iran are near 42 km [13,14]. Crustal thicknesses beneath the Zagros range from 35 km to over 60 km just across the plate boundary at the northeastern edge of the Sanandaj-Sirjan Zone [15–22]. The Alborz Mountains have a 55 km root beneath them [16,23], and the oceanic southern Caspian crust is 20–40 km-thick [13,24,25], which is unusually thick for ocean basins. It is thought to have formed during the Jurassic as a back-arc basin.

The best images of the crustal structure of Iran come from ambient noise studies [26,27]. Those studies show a slow Arabian crust, particularly in the uppermost crust, and a higher velocity Iranian crust. In contrast, at depth, the Arabian mantle is fast while the Iranian crust is slow. Pn velocities also show a fast Arabian mantle with velocities around 8.2 km/s and a slower Iranian mantle with velocities about 7.9 km/s [28]. Northwest Iran shows particularly low Pn velocities that correspond to the volcanism in that part of the middle east. Lg, Sn, and Pn are all attenuated in the middle east but not on the Arabian plate [29,30]. The southern Caspian Sea shows higher velocities consistent with its origin as an oceanic basin [24,29,30]. Of particular interest is the observed blockage of Lg and Sn for many paths [31–34].

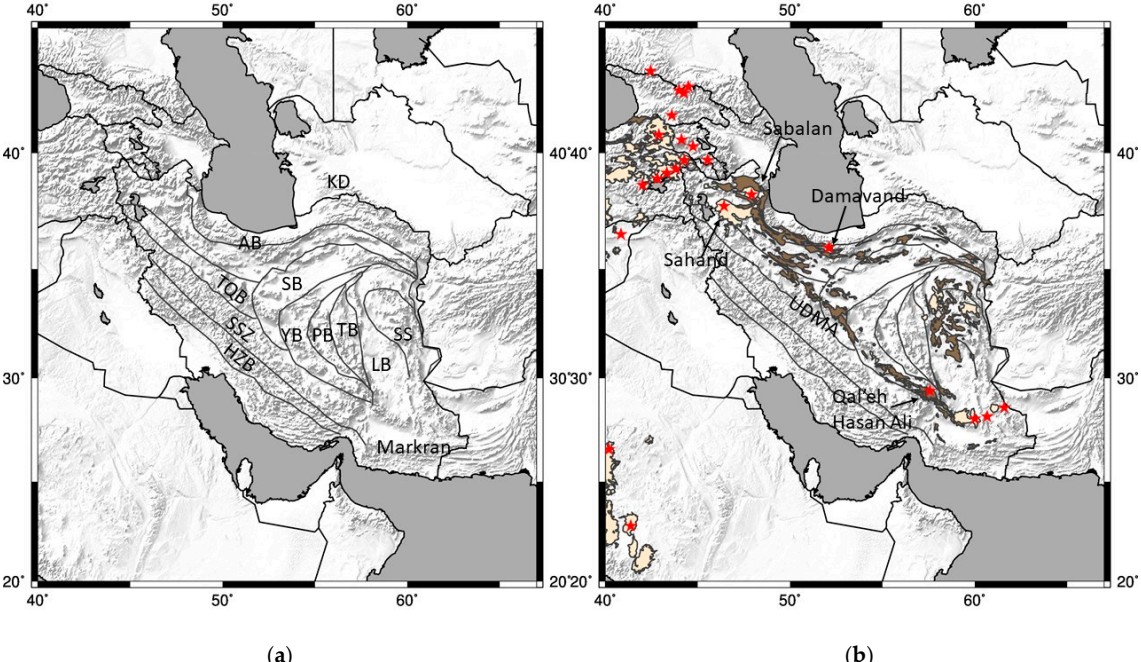

(a)  (b)

**Figure 1.** (**a**) Tectonic map of Iran with major blocks noted. The main plate boundary lies between the High Zagros Block and the Sanandaj-Sirjan Zone. (**b**) Red outlines are Neogene volcanics and tan outlines are Paleogene volcanics. Red stars are Holocene volcanoes. Faults from Arefifard [35]; volcanoes and volcanic outlines from Seber et al. [36]. AB—Alborz Belt; HZB—High Zagros Block; KD—Kopet Dagh; LB—Lut Block; PB—Posht-e-Badam Block; SB—Sabzevar Block; SSZ—Sanandaj-Sirjan Zone; SS—Sistan Suture (East Iranian Ranges); TB—Tabas Block; TQB—Tabriz-Qom Block; UDMA—Urumieh Dokhtar Magmatic Arc; YB—Yazd Block.

This paper uses amplitude, phase travel time, and group travel time data from the Iranian seismic network to image the attenuation, phase, and group velocities of Lg, Pg, Sn, and Pn waves in Iran. Phase travel times are the routine time measurements made at the beginning of the phase and used for event location. Group travel times are the times of maximum amplitudes. The terms phase velocity and group velocity are extended from their normal use with surface waves to mean the speed derived from the equations of motion and the speed of the wave envelope. Similar attenuation tomography has been conducted for the $M_L$ amplitudes in Iran using handpicked data by Maheri-Peyrov et al. [37]. They found that the maximum amplitude is from Lg, except for the southern Caspian Sea and Zagros where Sn dominates. Lu et al. (2012) [28] imaged Pn velocity structures beneath Iran and showed a clear contrast between the fast Arabian plate and slow Iranian plate.

Amplitude reductions with distance occur due to three processes: geometrical spreading, intrinsic absorption attenuation, and scattering attenuation. Geometric spreading is simply the change in amplitude due to wavefront spreading and the conservation of energy. For cylindrical geometric spreading, we expect a coefficient of $1/2$, a coefficient of 1 for spherical spreading, and a coefficient of around 2 for head waves [38] (p. 212). Intrinsic absorption refers to the frictional dissipation of energy in the media and scattering attenuation refers to the loss of energy due to scattering from inhomogeneities that may include deflections of the Moho, the surface, and other structural discontinuities. Both absorption and scattering attenuation are combined into the traditional concept of $Q$, with high $Q$ values representing low attenuation, where $1/Q$ is the fractional rate of energy loss per radian. $Q$ separates into intrinsic attenuation and scattering attenuation as follows: $Q_T^{-1} = Q_s^{-1} + Q_i^{-1}$, where $Q_T^{-1}, Q_s^{-1}$ and $Q_i^{-1}$ are the total, scattering, and intrinsic values for attenuation.

The group travel times are closely related to the peak delay time, which is defined as the time from the phase arrival time to its maximum amplitude time [39–41]. The significance of the peak delay time is that it is sensitive to scattering but insensitive to intrinsic attenuation. Briefly, intrinsic attenuation causes the entire phase, including its coda, to attenuate, while scattering attenuation causes the beginning of the phase to attenuate while pushing energy back into the coda, thereby increasing the peak delay time. The use of the delay time to identify regions of scattering has been performed in Japan [42–45], Isu-Bonin Arc [46], southern Aegean [47], and the Pyrenees [48]. Since the group travel time is the phase travel time plus the peak delay time, it can be interpreted as being due to both variations in phase velocity and scattering. Similarly for the group velocities, phase travel times have been widely used in seismology to investigate Earth, and the Iranian Seismological Center has picked and identified these for all crustal phases. Phase velocities within the crust are generally related to composition, while phase velocities in the mantle are primarily related to temperature.

This paper is focused on the comparison of tomography results for the four different regional phases with three differing types of tomography. Crustal velocities are compared to mantle velocities; P-velocities are compared with S-velocities. A comparison of phase velocities with attenuation allows the identification of partial melt; the comparison of group velocity with phase velocity and attenuation allows the identification of regional scattering variations. Regional geology variations are, of course, found to underlie variations in all seismic properties of the crust and mantle.

## 2. Data

Data are from the Iranian Seismological Center from January 2006 to August 2021 and are available for download at irsc.ut.ac.ir/istn.php. The seismic network uses Kinemetrics SS-1 velocity transducer seismometers (1 s period), Trillium medium band seismometers, and CMG and Trillium broadband seismometers. Magnitudes are computed using a modified Nuttli magnitude scale [49,50] (irsc.ut.ac.ir/istn.php). A complete seismic bulletin is kept with phase identifications, arrival times, amplitudes, and periods for over 70 stations. The average period is 0.35 s, so results in this paper are at about 3 Hz. Station locations may

be found in Figure S3 where they are shown as station delays. In this study, only vertical instruments were used for amplitudes and group travel times, vertical components were used for Pg and Pn phase velocities, and horizontal components were used for the Sg and Sn phase velocities.

The Iranian data, unlike most other magnitude datasets, also kept the time of the maximum amplitude arrivals, which are herein called group travel times. These are plotted vs. distance in Figure 2. Clearly, the magnitude maximum amplitude picked from different phases. There are four branches with apparent velocities of approximately 3.2, 4.6, 5.6, and 8.0 km/s. These are Lg, Sn, Pg, and Pn group arrivals. These group velocities are less than the phase velocities. Each data set was winnowed by fitting a line to it by eye and then drawing boundaries to include the phase. These are shown on Figure 2. Lg paths have an intercept of 6 s and velocities between 2.78 and 3.72 km/s. Pg paths have an intercept of 2 s and velocities between 4.99 and 6.40 km/s. Sn paths have an intercept of 17 s and velocities between 4.16 and 5.08 km/s. Pn paths have an intercept of 10 s and velocities between 7.09 and 8.70 km/s.

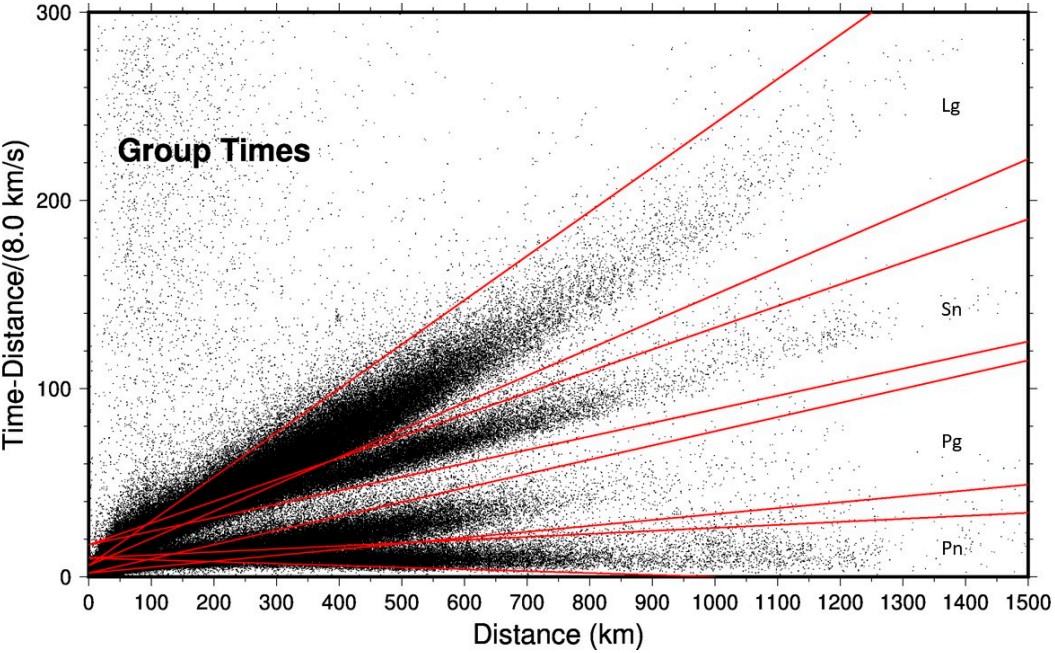

**Figure 2.** Group travel times. Four phases are recognized and winnowed by the corresponding sectors.

Outliers in these data sets are of key concern for any tomographic work. Exceptionally large outliers in amplitude and group velocity are already removed by the sector boundaries of Figure 2. In addition, during the initial data fit, all outliers greater than 2 magnitude units or 10 s are iteratively removed. All stations and events require at least two arrivals. There is a concern about the accuracy of the phase identification for the phase travel time picks. For Sg, arrivals beyond 100 km were used. This arrival is essentially the beginning of Lg, which is generally the largest arrival and thus readily identified. Pg data from beyond 25 km were used. Most Pg are first arrival data in this range and are easily identified. Sn arrivals past 100 km were used. This secondary phase is a particularly difficult phase to pick well and, as shall be seen, does not perform well at the phase pick tomography. Pn arrivals past 100 km were used. Most Pns are past the cross over distance and are first-arrival picks. Even when Pn is a secondary arrival, it is often visible and distinguished from Pg's first arrival.

### 3. Method and Initial Results

Travel-time tomography was performed for both the group travel-times and associated phase travel times [51]. The first step in tomography is to fit a line to the data to obtain the average velocity:

$$t_{ij} = a + d_{ij}s. \tag{1}$$

where $t_{ij}$ is the travel time between station $i$ and event $j$, a is the intercept, $d_{ij}$ is the distance, and $s$ is the slowness (inverse velocity). The next step is to invert the residuals using the following equation.

$$\Delta t_{ij} = a_i + b_i + \sum d_{ijk}s_k \tag{2}$$

The sum is over all cells k that the ray traverses. $\Delta t_{ij}$ is the travel time residual between station $i$ and event $j$, $a_i$ is the relative station delay, $b_j$ is the relative event delay, $d_{ijk}$ is the raypath distance in cell $k$, and $s_k$ is the slowness (inverse velocity) perturbation in cell $k$. The sum is over all cells the raypath traverses. In raytracing, the travel times are adjusted for depth and elevation. for the inverted quantities are the station and event delays and the slowness.

Inverting the amplitudes for attenuation variations works the same way as the travel time's inversion, with the exception that geometrical spreading must be accounted for [1–3]. The first step that needs to be performed is to fit the basic amplitude equation to graphs of log(amplitude)-magnitude vs. distance to determine the spreading value and an average $1/Q$ value. This equation, after taking base 10 logarithms, is as follows.

$$\log(A_{ij}) - M_j = a - K \log r - \left(\frac{1}{Q}\right)\log(e)\pi r_{ij}/vT \tag{3}$$

Here, $A_{ij}$ is the amplitude for event $i$ and station $j$, $M$ is the magnitude, $r$ is the epicentral distance, $v$ is the seismic group velocity, and $T$ is the period. The unknowns are the intercept, $a$, the spreading coefficient, $K$, and the inverse $Q$ quality factor.

The basic tomographic equation for amplitude inverts the log amplitude residuals and is given by the following.

$$\Delta \log(A_{ij}) - M_j + K \log r_{ij} = a_i + b_j - \sum \left(\frac{1}{Q}\right)_k \log(e)\pi r_{ijk}/vT \tag{4}$$

Here, $i$ is the station index, $j$ is the event index, and $k$ is the index to sum over cells in the raypath. $r_{ijk}$ is the distance the raypath takes in cell $k$. The quantities inverted for are the relative station and event gains, $a_i$ and $b_j$, and the inverse of the quality factor $Q$ in cell $k$.

In this paper, a grid of $1/4$ degree by $1/4$ degree cells was used for tomographic solutions. Rays were traced on an elliptical earth; cells were weighted by their area to cancel the effect of changing cell size with latitude. All inversions used Laplacian damping for smoothing; however, the delays and gains were not damped at all. All data were iteratively winnowed so that each station and event has at least two arrivals used in the inversion. This causes a large amount of noise to creep into the undamped station and event delays but provides maximum coverage. Attempts were made at using the picked period in the inversion, but they are small and in the denominator, so any error in this pick results in a poor fit. Replacing the group velocity by the distance divided by travel-time in the tomography produced similar maps but they had a higher rms from the inversion.

In none of these inversions were focal mechanisms considered since they are unknown. For Lg, it is known that the amplitude is relatively insensitive to the mechanism since the Lg wavetrain is composed of rays emanating from multiple parts of the focal sphere [52]. It is not known what the effect of focal mechanism is on the amplitudes of the other phases but it is considered noise in this paper. Other sources of noise are raypath focusing and raypath bending. Long-period surface wave studies have found focusing to be appreciable [53], but it is not clear how body waves focus; however, if phase velocity variations are relatively

small, this should not be a large effect. Raypath deviations due to phase velocity variations cause velocity variations to be misplaced, thereby blurring the image.

Plots of amplitude versus distance for the Iran data produced high spreading values of above 2.7 for spreading for all four phases and extremely high $Q$ values that sometimes even went negative. This is unphysical and a result of the use of velocity sensors where the measurement is at the maximum velocity of the seismometer's mass. Evidently, the maximum mass velocity squared is not a good indicator of wave energy nor of maximum ground velocity squared. It is not possible to extract the average spreading coefficient and average $Q$ from these data. Instead, a correction to recalibrate the velocity amplitudes back to $M_L$ displacement amplitudes is developed. I first fit a magnitude formula for data in the range of 150–1500 km using all phases and the MATLAB "robustfit()" fitting function:

$$\log v - M_N = -2.96 \log r + 0.000128r + 6.96 \tag{5}$$

where $M_N$ is the original Nuttli magnitude given by the array, $r$ is the distance in kilometers, and $v$ is the maximum instrument velocity that is measured. Thus, by using these data, a new station magnitude can be obtained.

$$M_N = \log v + 2.96 \log r - 0.000128r - 6.96 \tag{6}$$

The coefficients are different than that of the Iranian Seismological Center and that of [50] because a longer distance range has been chosen and an attenuation term was included (Table 1). For a local $M_L$ magnitude scale, there are several to choose from with, unfortunately, differing values. I chose to use the original $M_L$ scale for Southern California as it was in the middle and is somewhat of a standard scale (Table 1) [54,55].

$$M_L = \log A + 1.11 \log r + 0.00189r - 2.09. \tag{7}$$

**Table 1.** Regional magnitude relationships. Velocity refers to the maximum vertical velocity output of the SS-1 seismometer; displacement refers to the maximum displacement on horizontal Wood-Anderson instruments.

| Authors | Region | Amplitude Type | Spreading Coefficient | Attenuation Coefficient | Constant Term |
|---|---|---|---|---|---|
| This study | Iran $r > 150$ km $r < 1500$ km | Velocity | 2.95 | −0.000150 | 6.98 |
| Iranian Seismological Center | Iran $r > 106$ km $r < 600$ km | Velocity/$4\pi$ | 2.5 | - | −1.8 |
| Rezapour 2005 [50] | Iran $r > 170$ km $r < 1000$ km | Velocity/$4\pi$ | 2.6 | - | −2.2 |
| Askari et al., 2009 [56] | Central Alborz | Displacement | 1.1725 | 0.0021 | −0.4450 |
| Rezapour and Rezaei 2011 [57] | NW Iran | Displacement | 0.9252 (vert.) 0.9993 (horiz.) | 0.0030 0.0029 | 0.8496 0.7114 |
| Emami, Rezaei, and Rezapour 2014 [58] | NW Iran | Displacement | 1.4050 | 0.0019 | 3 |
| Shoja-Taheri et al., 2007 [59] | Iran $r < 96$ km $96 < r < 131$ km $r > 131$ km | Displacement | 1.01 −0.14 0.14 | 0.0002 0.0002 0.00020 | - |
| Bormann et al., 2013; Hutton and Boore 1987 [54,55] | Southern California <1000 km | Displacement | 1.11 | 0.00189 | −2.09 |
| Shoja-Taheri et al., 2008 [60] | NE Iran > 0 km $0 < r < 106$ $106 < r < 347$ $r > 347$ | Displacement | 1.37 1.38 0.597 0.415 | 0.002 0.0033 0.0033 0.0033 | - |

To correct velocity amplitudes to displacement amplitudes, these two expressions are set equal. Then, we have the following equation.

$$\log A = \log v + 1.85 \log r - 0.00202r - 4.87 \tag{8}$$

This equation effectively sets the average spreading and attenuation levels to those of the Southern California model. Amplitude data are shown in Figure 3 along with travel time data. All plots have coherent data clouds.

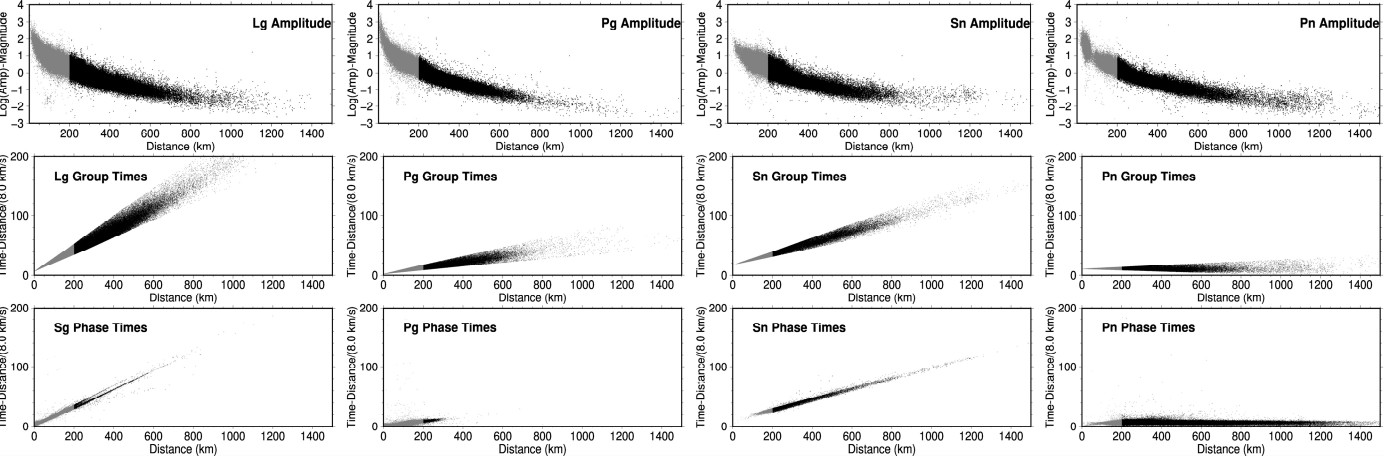

**Figure 3.** Raw data for travel times and amplitudes. Gray data were not used in the inversion. Amplitude and group velocity data at distances greater than 200 km were used for all four phases; Sg and Pg travel time picks for distances greater than 25 km were used; Sn and Pn travel time picks for distances greater than 100 km were used.

## 4. Results

### 4.1. Lg

The raypath maps are shown in Figure 4. If full raypaths are plotted, the figure becomes saturated; thus, only every 21st element of each raypath was plotted. This makes it difficult to see individual raypaths but shows the coverage variations well. For Lg, very few paths cross the southern Caspian Sea and the few that do are probably stray Sn raypaths that crept into the data set. Lg is blocked by the transition from continental to oceanic paths [33,61].

The tomographic inversion for Lg attenuation is shown in Figure 5. Station and event delays and the appropriate checkerboard resolution tests for them and other phases may be found in Supplementary Figures S1–S4. We use checkerboard resolution tests in which we invert a synthetic data set, plus noise, for a set of square anomalies alternating between low and high. The smallest set of squares that can be imaged shows the minimum resolution length. For all phases, squares of 2 degrees are well resolved. Arrivals numbering 132,040 were used in this inversion with a minimum of 2 arrivals per station and per event. Maximum residuals were kept under 2 magnitude units. The spreading value was 1.21 with an average $Q$ of 694. The rms was 0.16 magnitude units. Errors for the average $Q$ and average spreading are exceedingly small (order of 10 and 0.01) because tens of thousands of data points are used to invert for only three data points. Lg has significant areas of both low and high attenuation. The Lut Block and the Tabas Block form a low attenuation pair. There is lower attenuation on the Arabian Plate. The Kopet Dagh also shows low attenuation. In the north, a good portion of the Alborz Mountains has high attenuation. Rahimi et al. [62] also observed high attenuation there. The NW-SE swath of high attenuation through central Iran correlates with the Eocene Urumeih-Dokhtar Magmatic Arc. This was the arc associated with the northeast subduction of the Neotethys Ocean prior to the collision of Arabia with Asia. There is a high attenuation anomaly in northern Sistan.

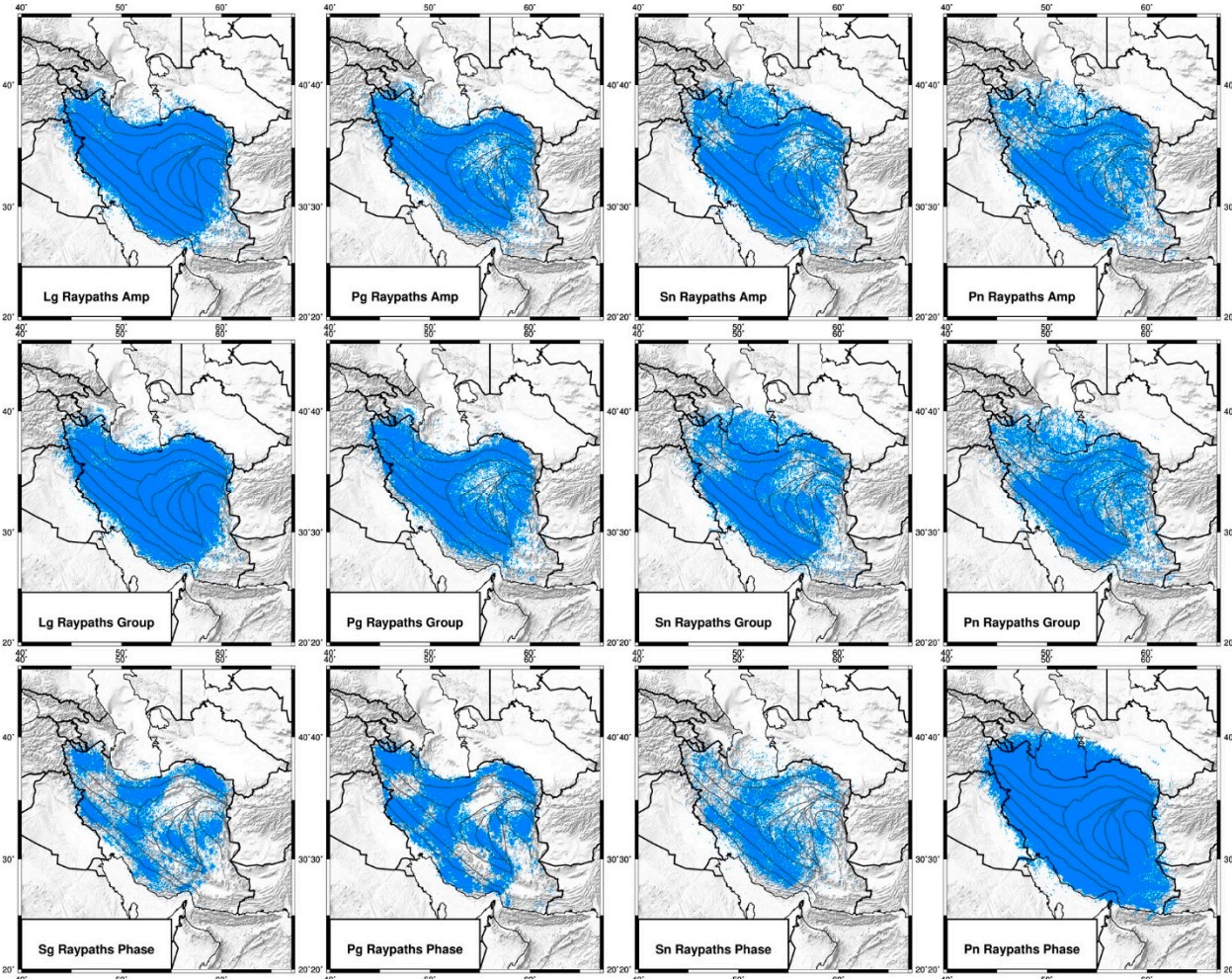

**Figure 4.** Raypaths. Drawing complete raypaths saturates the images, s only every 21st element of each raypath is plotted. While this obscures individual raypaths, it shows the coverage well. Note the lack of Sg, Lg, and Pg raypaths beneath the southern Caspian Sea.

The spreading rate for Lg, 1.21, and its *Q* value of 694 can be compared to others in the Middle East. A theoretical study has found an Lg spreading of 0.83 [63]. Akinci et al. [64] found that spreading is 0.6 to 0.9 in western Anatolia and 0.5 in southern Spain with *Q* values of $82 \times f^{1.0}$ and $83 \times f^{0.88}$ (246 and 215 at 3 Hz). Mahood and Hamzehloo [65] observed a *Q* of $59 \times f^{1.00}$ (177 at 3 Hz) for S-waves with an assumed spreading coefficient of 1 for east–central Iran. Both the spreading rates and the *Q* values in these studies are lower than found with the Iran bulletin data. For the southern Alborz region, Naghavi et al. [66] observed Lg *Q* values of $267 \times f^{0.71}$ (582 at 3 Hz), which are very close to the *Q* values found in the present study assuming a spreading of 0.5. Furthermore, their attenuation map closely aligns with Figure 5. Meghdadi and Shoja-Taheri [67] found a *Q* of $166 \times f^{1.13}$ (574 at 3 Hz) and a spreading rate of 0.73 for Iran. The Sabalan volcano has extremely high attenuation [62]. These latter values for *Q* are more in line with the results of the present study. Differences occur because the spreading and *Q* values were adjusted to be those implied by Southern California magnitude formula 6 and different studies make differing assumptions about spreading. This formula is also more in line with those in Table 1. Thus, there is an inherent tradeoff between the absolute values of spreading and $1/Q$.

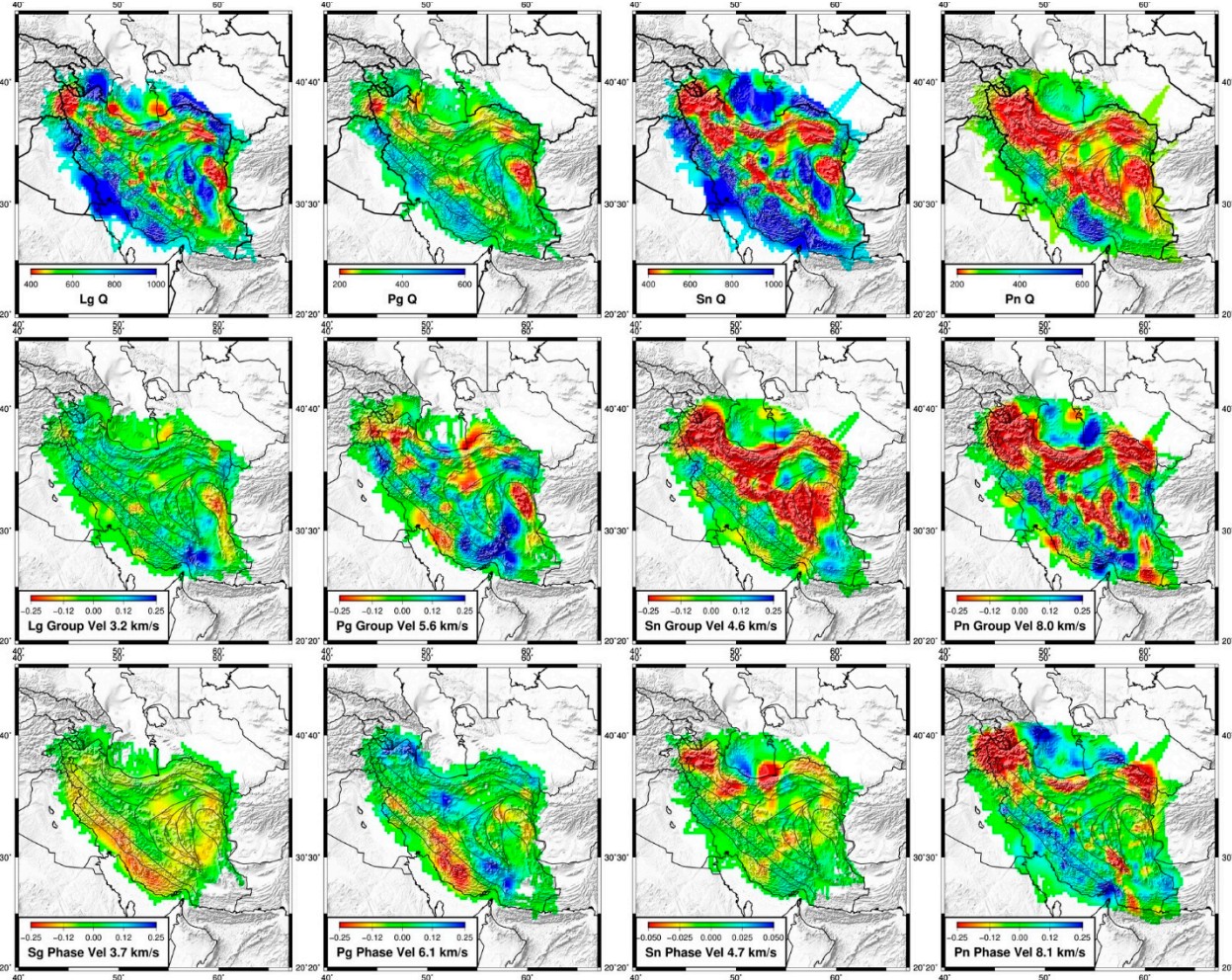

**Figure 5.** Velocities and attenuation. Top plots are the attenuation, middle plots are of group velocity, and bottom plots are of phase velocity. The color bar is stretched to produce equal contours in $1/Q$. Note the reduced velocity scale for Sn phase arrivals.

Lg attenuation for the middle east was mapped by Pasyanos et al. [30], and the results are very similar to this study. In particular, the Urumieh Dokhtar Magmatic Arc, the northern Sistan Suture anomaly, and the Aborz are all mapped as high attenuation. Furthermore, the maps of Pasyanos et al. show that the very high attenuation in the northwest Alborz is part of a larger anomaly covering Armenia and Azerbaijan. This anomaly is associated with volcanism in the area. In another study, Pasyanos et al. [29] mapped crustal and mantle attenuation for both P- and S-waves for the greater Arabian Peninsula. They find that Lg attenuation is lower on most of the Arabian plate but is slightly higher in the Zagros region and higher on the Iranian Plate. Kaviani et al. [34,68] mapped Lg $Q$ throughout the middle east using the two-station method but were not able to resolve the same features found with bulletin data.

Group velocity tomography was also performed using the times of the maximum amplitudes. Distances from 200 to 1500 km were used with a minimum of two arrivals at each station and event, and residuals were limited to 10 s. Arrivals numbering 136,112 were used, and the rms is a large 3.1 s. Despite the noise, checkerboard tests showed that two-degree squares could be imaged, although the station and event delays are noisy (Figure 6, Figures S3 and S4). The average group velocity was 3.2 km/s. A major low group velocity anomaly is in the northern Sistan Suture. Maheri-Peyrov et al. [37] also imaged the group velocity of the maximum amplitude only using Wood-Anderson seismometer

equivalents. They found that it was often confused with Sn. Similarly to the present study, they observed low group velocities beneath the Urumieh Dokhtar Magmatic Arc and no Lg propagation across the southern Caspian.

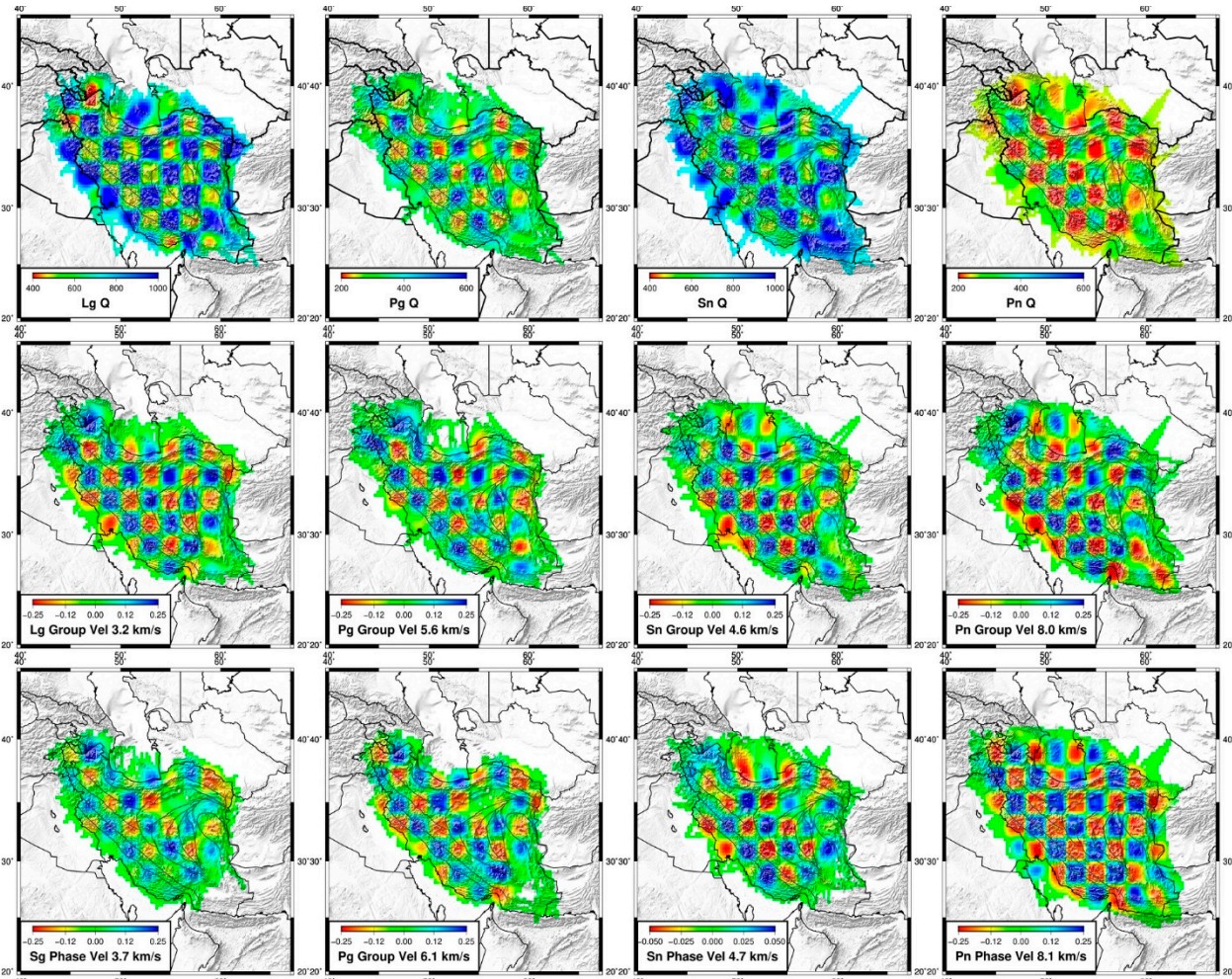

**Figure 6.** Checkerboard tests of velocity and attenuation. The original checkerboard had attenuation and velocity squares of two-by-two degrees. The station and event delays also alternated every two degrees. Representative noise was added. Velocity squares are ±0.2 km/s and attenuation squares are ±150.

There are no phase velocity measurements for Lg; however, we can use the phase picks for Sg to image S wave phase velocities for comparison. Sg generally arrives right before Lg and is never the first arrival. It is picked, in this data set, out past 600 km and may well be the Lg initial arrival at those distance ranges. The inversion used 107,980 arrivals in the 25 to 1500 distance range with a requirement that each station and event has at least two arrivals, although most Sg arrivals are at shorter distances. The average Sg velocity is 3.7 km/s, and the rms is 0.51 s. Figure 5 shows the results of the tomography. Checkerboard tests show that the tomography can resolve velocity variations to better than 2 degrees (Figure 6). Zagros has a clear low-velocity signature; however, raypath coverage (Figure 4) is poor for the rest of Iran. Similarly to Lg, Sg does not cross the southern Caspian Sea (Figure 4). The most obvious feature is the change in velocity across the plate boundary. The image obtained here for Sg looks remarkably similar to the 30 km deep image from ambient noise tomography [26,27].

*4.2. Pg*

Not many Pg raypaths are long enough to cross the southern Caspian Sea and those that are may be Pn that leaked into the data set. Coverage is much better for the Pg amplitude and group velocity data sets than for the phase picks.

I inverted the amplitudes, maximum times, and travel times of Pg for attenuation and velocities. The attenuation map (Figure 3) shows a pattern similar to that of Lg arrivals. This is unsurprising since they both sample the crust. The spreading factor for the Pg was 0.97 and the $Q$ was 314. There were 38,305 arrivals. The rms is 0.14 magnitude units. A two-degree checkerboard test can be resolved (Figure 6). As with the Lg attenuation, in the northern Sistan Suture, there is again a high attenuation anomaly that contrasts with the low attenuation of the Lut and Tabas Blocks to the southwest of it. There is lower attenuation on the Arabian Plate. There is high attenuation in the eastern and western Alborz and low attenuation in the Kopet Dagh and the very northeastern Alborz. These features were also present in the Lg tomography. Again, we have the Urumieh Dokhtar magmatic arc cutting NW-SE through the center of the figure.

A theoretical estimate of Pg spreading is 1.5 [61]. An estimate of $Q$ for Pg of $54.2 \times f^{1.0096}$ (150 at 3 Hz) by Bajestani et al. [69] for all of Iran is about half that of the bulletin data. He obtained Pg spreading factors of 1.2 within 90 km of the source; however, this factor becomes negative distances from 90 to 175 km due to reflections from the Moho. He also provides a table of Pg and Pn Q estimates from around Iran and the world, and they correspond well to this low $Q$. Motaghi and Ghods [70] found that $Q = 109 \times f^{0.64}$ (220 at 3 Hz) for the central Alborz. However, in both these studies, Pg amplitudes were measured at local distances and not the regional distances of over 200 km that are considered with the Iranian Bulletin. Apparently, this makes a difference for Pg.

The map of seismic group velocity for Pg uses arrivals from 200 to 1500 km distances (Figure 3). There are 41,858 arrivals, the average group velocity is 5.6 km/s, and there is a large rms of 2.4 s. The checkerboard test shows that two-degree cells are resolved (Figure 6) and that the station and event delays are very noisy (Figures S3 and S4). The map of Pg group velocity from the travel times shows a slow velocity feature on the Arabian Plate, and the northern Sistan Suture and northeastern Alborz mountains have low velocity. A large high group velocity zone exists in southern Iran in the Lut block.

For Pg phase velocities, I used the distance range of 25–1500 km, although most arrivals are from less than 150 km where Pg is usually the first arrival. There were 536,139 travel times used, and these produced an average velocity of 6.1 km/s and an rms value of 0.50 s. Checkerboard tests with this amount of noise show that two-degree cells can be imaged (Figure 6). The map shows that low velocities are clearly associated with the Zagros and northern Sistan Suture.

Maheri-Peyov et al. [71] also used Pg arrivals from Iran but in a three-dimensional inversion. They clearly imaged the low velocities of the Arabian Plate and show that these extend deep into the crust; however, they obtained a lower phase velocity in northeastern Alborz that the Iran Bulletin data do not image. Similarly, Rezaeifar and Kissling [72] performed three-dimensional crustal tomography and found lower velocities on the Arabian Plate and higher velocities in the interior. There have been several studies of Pg that focused on the main volcanoes of Iran: Sahand, Sabalan, and Damavand [71,73–76]. They all observed low-velocity zones in the crust near the volcanoes but these features are small and generally beneath the resolution provided in this tomography study.

*4.3. Sn*

Similarly to Lg and Pg, I inverted the amplitudes, group travel times, and phase travel times of the Sn picks. The average spreading is 1.69 with a $Q$ of 733. The rms of the residuals is 0.18 magnitude units. There are 62,169 raypaths and two-degree checkerboards can be resolved (Figure 6). The amplitude plot shows that the Alborz and the northern and southern Urumieh Dokhtar Magmatic Arc have high attenuation. The Sistan Suture shows up clearly. The southern Caspian Sea and the southern Zagros show very low attenuation.

Sn attenuation has also been imaged by Pasyanos et al. [29]. Those results show very low attenuation on the Arabian Plate and higher attenuation on the Iranian Plate with values averaging 562 in the 1–2 Hz passband.

Group travel times for were inverted for Sn group velocity variations using the 200–1500 km distance range. The average group velocity is 4.6 km/s. There were 71,543 arrivals with a very large rms of 2.1 s. Nevertheless, the two-degree checkerboard test says that we can resolve to that level (Figure 6). The largest anomaly is the low group velocity of the northwest and central Alborz. The northern Sistan Suture anomaly is apparent, as is the Urumieh Dokhtar Magmatic Arc. Sn propagates across the southern Caspian Sea but there is a paucity of data relative to northwest and eastern Iran.

The phase velocity map for Sn was made for distances of 150–1500 km. It suffers from having only 19,864 raypaths in it. The mean velocity is 4.7 km/s and the rms is 0.52 s. Surprisingly, the maximum Sn phase velocity anomaly was only about 0.05 km/s and the map had to be plotted on a different scale. Pei et al. [77] found Sn velocity anomalies in Iran about double that and they were extended throughout the country. The checkerboard test shows that the image should be able to have good resolution and the rms is reasonable. However, Sn is a particularly hard phase to pick consistently and is often absent in Iran [32,33]. It must be concluded that the picks for Sn are poor and produce a flat tomography image; however, the mean Sn velocity is well determined and there are some low anomalies in the Sn velocity image in western and central Alborz.

*4.4. Pn*

Pn amplitudes were inverted for distances of 200–1500 km. There were 32,537 paths, an rms of 0.19 magnitude units, a spreading of 0.99, and a $Q$ of 254. Pn paths were far less numerous, so two-degree checkerboard tests are weakly resolved. Nevertheless, the southern Caspian Sea clearly shows low attenuation as does most of the Zagros. High attenuation zones are beneath the Alborz mountains, the northern Sistan Suture anomaly, and the southeastern Urumeih-Dokhtar Magmatic Arc. Similarly to Sn arrivals, Pasyanos et al. [29] also show that Pn attenuation is low beneath the Zagros and higher beneath Iran. My spreading factor agrees with that of 1.11 at 1 Hz found by Zhu et al. [78] for Eastern Canada; however, their $Q_{pn} = 189 \times f^{0.87}$ (491 at 3 Hz) is higher. Bajestani et al. [69] found a spreading factor of 1.3 and $Q_{pn} = 306.8 \times f^{0.51}$ (535 at 3 Hz) in Iran.

Pn group travel times were inverted for group velocities. There were 35,148 raypaths, and the mean Pn group velocity is 8.0 km/s with a large rms error of 1.8 s. The checkerboard test showed that two-degree cells can be resolved; however, the undamped station and event delays are extremely noisy. The southern Caspian shows high velocities associated with the oceanic sea; central Alborz had slow velocities. The Urumieh Dokhtar Magmatic Arc shows low Pn group velocities. To a large extent, the group velocities mimic the phase velocity but more intensely.

For Pn phase times, distances were from 100 to 1500 km where Pn is the first arrival, a minimum of two arrivals at each station and event, and a maximum delay of 10 s. At the minimum distance used, Pn may be a secondary arrival due to the thickness of the Iranian crust; nevertheless, the analysts did identify the arrivals as Pn. This results in 498,525 raypaths with a mean velocity of 8.0 and an rms of 0.77 s. Unlike Sn, there are many structures in Pn tomography. Pn phase travel time velocities yield a map that is basically an update to that of Lu et al. [28] and very similar to that of Amini et al. [79], although they have larger velocity variations than found here. There are high velocities in the Arabian plate and southern Caspian Sea. The Urumieh Docktar Magmatic arc comprises low velocity. The Alborz Mountains show low velocities in the eastern, central, and western parts. Notice that Pn propagates everywhere.

## 5. Geophysical Interpretation

P-waves show higher attenuation than S-waves for both crust and mantle with a ratio of two to three while S-waves show higher spreading values. Figures from

Pasyanos et al. (2021) [29] also show that P-wave attenuation is generally higher, as do other results for regional phases [65,80,81]. Gusev and Abubakirov [82] estimate that $Q_s/Q_p = 1.9$ for Kamchatka. Average spreading and $Q$ values are biased to the southern California velocity model.

The group velocity for body waves is a new parameter and needs to be explicitly related to the peak delay time. The phase travel time may be expressed as $t_p = s_p x + a_p$ where $s_p$ is the phase slowness and $a_p$ is the phase intercept. Similarly, the group travel time is $t_g = s_g x + a_g$. Taking the difference of these gives the peak delay time $\Delta t = t_g - t_p = (s_g - s_p)x + (a_g - a_p) = s_x + (a_g - a_p)$, where $s = s_g - s_p$ is the peak delay time slowness, a function of scattering and not intrinsic attenuation. This result contrasts with the power law formulation commonly used to express the peak delay time as a function of distance but is required by the observation of constant group velocity [42,43], [39] (p. 371), [45,47,48]. The intercept, $a_g - a_p$, is the difference between the group velocity intercept and the phase velocity intercept and represents the peak delay time of the source time function. From Table 2, these are less than four seconds even though the numbers are from different data sets. The group slowness is related to the phase slowness and peak delay slowness by $s_g = s_p + s$. Thus, group velocities depend on both the phase velocity and the amount of scattering, with low group velocities indicating low phase velocities or increased scattering. In the figures, lower crustal phase velocities and higher mantle phase velocities on the Arabian plate influenced group velocities, especially Pn group velocities.

**Table 2.** Results from inversions.

| Phase | Phase Velocity | Group Velocity | Spreading | Q | Phase Intercept | Group Intercept |
|---|---|---|---|---|---|---|
| Lg | 3.7 km/s (Sg) | 3.2 km/s | 1.21 | 694 | 3.96 s (Sg) | 2.11 s |
| Pg | 6.1 km/s | 5.6 km/s | 0.97 | 314 | 1.8 s | 1.7 s |
| Sn | 4.7 km/s | 4.6 km/s | 1.69 | 733 | 12.7 s | 19.9 s |
| Pn | 8.1 km/s | 8.0 km/s | 0.99 | 254 | 8.6 s | 11.9 s |

A comparison of average group velocities to the average phase velocities and attenuation can be used to separate the effects of intrinsic attenuation and scattering attenuation. Note that almost all regions with low group velocities correlate with either high attenuation or low phase velocity. Slower group velocity means more scattering and/or lower phase velocity. If there is high attenuation with low group velocity but the phase velocity does not change, we can conclude that scattering dominates the attenuation. If there is high attenuation without a low group or phase velocity anomaly, then we can conclude that intrinsic absorption dominates the attenuation. The group velocities for the two crustal phases, Lg and Pg, are 0.4 to 0.5 km/s lower than their respective phase velocities while the group velocities for the two mantle phases, Sn and Pn, are only 0.1 km/s lower than their phase velocities. This indicates that the mantle scatters less than the crust even though its total attenuation is comparable.

The potential for partial melt exists within the Iran crust, particularly beneath the main volcanoes. However, in the crust, the only low phase velocities we see are associated with Zagros sediments. There are no low crustal phase velocity anomalies associated with the Alborz or UrumiehDokhtar Magmatic Arc. This suggests that the strong attenuation observed in Lg and Pg was due to high attenuation in the intrusive volcanic rocks and that partial melt in the crust is not required. There remains the possibility that scattering in the mantle contributes to their attenuation, particularly beneath the Sanandaj-Sirjan Zone and Alborz where there are crustal roots, but neither region shows much in the way of low group velocities that would indicate such scattering.

The Iran bulletin data show that average mantle phase velocities, Sn and Pn, are high at 4.7 and 8.1 km/s. Pn velocities reach as low as 7.8 km/s beneath the Urumieh Dokhtar Magmatic Arc and even lower under parts of the Alborz, including Holocene Sahand, Sabaland, and Damavand volcanoes. This certainly indicates a hot mantle with perhaps

partial melt. Another volcano of interest is the Holocene Qal'eh Hasan Ali maars. It is at the southern end of the Urumieh Dokhtar Magmatic Arc, but it is not considered to be part of the arc [83]. This shows up clearly as low velocity on the Pn phase and group velocity maps as well as a zone of high attenuation. The Markran volcanoes to the southeast do not show up at all.

Iran is known for its blockage of Lg and Sn waves [31–33]. Clearly, the Iran Bulletin amplitude data presented here refute these observations, except the southern Caspian where Lg raypaths do not cross (Figure 4). Furthermore, Pasyanos et al. [29,30] and Kaviani et al. [34,68] were clearly able to measure these phases; thus, they are not altogether extinguished. However, the poor quality of the Sn phase picks does indicate how difficult the phase is to pick. Surface waves indicate a slow upper mantle that may be in a state of partial melt with a negative S-wave velocity gradient [84,85]. However, this slow uppermost mantle cannot extend to the Moho in Iran as the average Sn and Pn velocities are high. Perhaps there is a thin, but positive, velocity gradient immediately beneath the Moho in a solid layer that can propagate higher frequency waves.

## 6. Geological Interpretation

The plate boundary shows a marked transition from a low-velocity Arabian crust in the southwest to a higher velocity Asian crust in the northeast in phase velocity maps. Only the Pg group's velocity map showed low group velocity. This suggests that scattering dominated the group velocity patterns in the crust. Both the Pn group velocity and phase velocity maps show higher velocities in the Arabian Shield mantle with very low attenuation. The high velocities of the Arabian Plate seen in both Pn and Sn extend several hundred kilometers down to form a thick Arabian lithosphere [86]. The observation that the boundary is more intense in the Pn group velocity map suggests that the Arabian Plate mantle has low scattering as well as low attenuation. The Zagros Mountains do not attenuate any of the phases despite its sedimentary thickness.

The Urumieh Dokhtar Magmatic Arc extends NW-SE across Iran from the northwestern Alborz to Makran and its effects can be seen as a high attenuation zone for all four phases. Work using ambient noise tomography in Iran also imaged the Urumieh Dokhtar Magmatic Arc as a high-velocity region in the uppermost crust [87]. This was the Eocene volcanic arc associated with the northeastern subduction of the Neotethys Sea beneath Iran prior to the Oligocene continental collision [5,11]. While primarily active in the Eocene, some volcanism continued until the Oligocene. Volcanic regions have been found to have high attenuation in other studies [3,37]. The lack of an associated low Sg or Pg phase velocity anomaly shows that no partial melt is associated with this anomaly in the crust; however, Pn velocities as low as 7.8 km/s show that the mantle is hot there. This observation is backed by that of Manaman et al., Amini et al., and Lu et al. [14,76,79]. The lack of a group velocity anomaly associated with the magmatic arc suggests that the anomaly is primarily due to intrinsic attenuation.

In the northern Sistan suture, a consistent low group velocity and high attenuation anomaly exists for all four phases. It is coincident with a gravity low that indicates a slightly thickened crust [88,89]. The Sistan Suture is basically one large suture zone region marked by flysch, mélange, and ophiolites [12,90]. It formed where the Cretaceous to Eocene north–south Sistan Ocean existed. Subduction for this ocean basin was at the northeast and ceased when the Lut block collided with it. However, the Lut Block is irregularly shaped with a promontory to the south. This left the ocean to the north unable to completely close and hence a different crust and mantle column developed there. Undoubtedly, burial, metamorphism, intrusion, and underplating have substantially modified the area with the flysch and Eocene intrusions contributing to the high attenuation. There is a slight low-velocity anomaly in the Pg phase velocity there, but it is not larger than that caused by the Zagros sediments and, hence, is unlikely to be due to partial melt. The large group velocity anomaly indicates the feature attenuates through heavy scattering.

The southern Caspian Sea is viewed as the remnants of a Jurassic back-arc basin [24,91]. It is well imaged in the Pn group and phase velocity maps and the Sn group velocity as a high-velocity region and low attenuation region. The raypath map of Lg and Sg (Figure 4) shows that only a few of these paths cross the Caspian Sea. Pg raypaths are not generally long enough to cross the sea. Thus, waveforms that cross the Caspian Sea have only Sn and Pn as regional phases.

The Alborz range from northeastern Iran to northwestern Iran with high attenuation in the Lg, Sn, and Pn images but only slightly higher in the Pg. This agrees with authors who have studied attenuation in the Alborz [62,92]. The Sg and Pg phase velocities beneath the central Alborz, just south of the Caspian Sea, are only slightly lower. There is a large low-velocity anomaly in the mantle beneath central Alborz that is apparent in the Sn and Pn groups' velocities as well as the Pn phase velocity, indicating a warm mantle. This is beneath Damavand volcano. The low mantle group velocities associated with these high attenuation zones suggest a high degree of scattering. The low mantle phase velocity anomaly in the western Alborz is associated with high attenuation, suggesting a hot mantle there. This region is beneath Sahand and Sabaland volcanoes. The low mantle group velocities coupled with the high attenuation again suggest scattering. As stated earlier, this high attenuation anomaly continues into Albania and Azerbaijan [30] and can be associated with Sabalan and Sahand volcanoes in Iran and those in Armenia and Azerbaijan. Eastern Alborz also shows low Sn and Pn phase velocities that accompany high attenuation, suggesting a hot mantle there, and the lack of a group velocity anomaly shows that there is no scattering.

## 7. Conclusions

The amplitudes and velocities of regional seismic phases beneath Iran correspond to wave type, regional tectonics, and geology. P-wave attenuation is higher than S-wave attenuation in both the crust and mantle. Spreading is higher for S-waves. The group velocity is related to both the phase velocity and the amount of scattering. Differences in the group and phase velocities show that the crust scatters more than the mantle despite similar attenuation levels. Clear geologic features imaged include the Urumieh Dokhtar Magmatic Arc, the Zagros Mountains and associated plate boundary, the Sistan Suture, the Lut and Tabas blocks, the Sistan Suture, southern Caspian Sea, and the Alborz. The plate boundary has low crustal velocity and crustal attenuation, with high mantle velocities and low mantle attenuation on the Arabian Plate. The lack of a low group velocity anomaly in the Arabian plate indicates that no scattering attenuation is occurring there. There is no sign of attenuation associated with Zagros sediments even though phase velocities are affected by them. High attenuation is not associated with sediments but with the igneous features of Iran. The Urumeih Dokhtar Magmatic Arc has high attenuation, showing that volcanic rocks have high attenuation even when they are solid, but there is no evidence of scattering there. The northern Sistan suture shows distinct anomalies in the attenuation and group velocities for all four phases. It is a closed ocean basin that has been affected by deposition and intrusion, causing both attenuation and scattering. The southern Caspian Sea does not allow Lg or Pg propagate. The main volcanoes of Iran, Sahand, Sabalan, Damavand, and Qal'eh Hasan Ali do not show up in the crust except for slightly high attenuation. Any partial melt in the Iranian crust is in small pockets. However, major volcanoes show up as low velocity and high attenuation features in the mantle. Those velocities, however, are still high, suggesting that the mantle anomalies are hot but with little partial melt.

**Supplementary Materials:** The following supporting information can be downloaded at: https://www.mdpi.com/article/10.3390/geosciences12110397/s1, Figure S1: Station delays and corrections; Figure S2: Event delays and corrections; Figure S3: Checkerboard tests of station delays; Figure S4: Checkerboard tests of event delays.

**Funding:** This research received no external funding.

**Data Availability Statement:** All data are from the Iranian Seismological Center at http://irsc.ut.ac.ir/ (accessed on 16 October 2022).

**Acknowledgments:** We thank the Iranian Seismological Center for producing this excellent data set. Data are from http://irsc.ut.ac.ir/ (accessed on 16 October 2022). All figures were made using GMT version 6.1 software by Wessel et al. [93].

**Conflicts of Interest:** The author declares no conflict of interest.

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
