# Peer review of "Two-Dimensional Attenuation and Velocity Tomography of Iran"

_geosciences, doi:10.3390/geosciences12110397_

Round 1

Reviewer 1 Report

This paper applies standard techniques to earthquake catalogue data to determining the propagation velocity and attenuation of crustal phases in the Iran region.  The data set  of arrival times and amplitudes is large and the author has a great deal of experience in this area.  I have just a couple of minor points.  Some of text could stand revising.  The caption to figure 2 is incorrect.  I assume the pairs of red lines in Figure 2 and show the how the phase is bounded.  Otherwise, the paper is ready to publish.

Author Response

I have fixed the caption to figure 2.

Author Response

Reviewer 2:

I thank this reviewer for a great job.  He has much more insight into Iran than I do.

Maheri-Peyrov et al. (2016) are already in the article on page 10 and as reference 62.  I also added this reference in line 93.  To be complete in the introduction, I also commented on Lu et al. (2012) in line 78.

Thank you for the added references.  I have used them all.  The ambient noise results are similar to what I get.

Figure 1:  It’s great to have a reviewer who’s very familiar with Iranian geography I got these names and the tectonic blocks from the referenced publication (Arefifard) at the beginning of the project.  I wish I had found better outlines.   I updated HZ to HZB on the map; got rid of the ZO (Zagros Orgeny).  I also combined CO (pre-cimmerian orgeny) and AB (Alborz) into just AB and moved it to the proper location.  I expanded my introduction with a paragraph on the seismic structure of Iran.

Data section comments: 

Station locations are actually shown in Figure S3, the map of the station delays.  I don’t want to lengthen the paper with another figure so I just put in a comment about this on line 146.

On line 141 (version with corrections) I corrected the list of instruments.

On line 161 I added a paragraph about how outliers are dealt with.   For amplitudes, outliers greater than 2 magnitude units are iteratively removed in the initial data fit.  For travel times, outliers greater than 10 seconds are removed.  Each station and event have at least two arrivals.  From experience, these parameters affect the coverage, but the tomographic patterns are stable.

The reviewer makes a very insightful comment about the data quality and especially the travel time pick quality and identification by the Iranian array.  In particular, he says that the Sg, Pg, Sn, and Pn arrivals are not well identified.  This explains why I get such small velocity variations for the Sn and I have changed my conclusion on this accordingly to state that the Sn picks are poor quality (Line 442).  For Sg, arrivals beyond 100 km were used.  This arrival is essentially the beginning of Lg which is generally the largest arrival.  Pg data from beyond 25 km were used.  Most the Pg are first arrival data in this range and are easily identified.  Sn arrivals past 100 km were used.  This is a particularly difficult phase to pick well, and, as shall be seen, does not perform well at the phase pick tomography.  Pn arrivals past 100 km were used.  Most of these Pn are past the cross over distance and are first arrival picks.  Even when Pn is a secondary arrival it is often visible.  I put the above in the data section starting on line 161.  Finally, the time-distance plots of Figure 3, third row, shows that all four phases are linear.  (Like all bulletins the Iranian bulletin has issues, but I’ve looked at many bulletins and this is the best bulletin being produced in the world.)

Equation 3.  Yes, it would be nice to add a multiplicative coefficient to the Mj terms.  However, that makes the inversion non-linear and I’m not set up to do this without investing a lot more time and I would not be able to get this revision back.  I did do this once through iteration, long ago, with Chinese data and got coefficients very near to one, so I quit playing with that term.  I note that any systematic shift in the Mj for event j, will be picked up as an event correction in equation 4.

Line 156 (now 210):  I changed that line to “All data is iteratively winnowed so that each station and event has at least two arrivals used in the inversion “. This means I did not use any events or stations that had only one recording associated with them.  Two arrivals is an admittedly low number, but it was picked to maximize raypath coverage.

Line 202:  I changed the line to indicate that the 2.7 spreading was for all four phases.  This number was used to recalibrate all the data back to Ml amplitudes but not interpreted.

Table 1: Askari and Ghods 1987 was a conference paper so I removed it.  I do have a copy.

Figure 3:  I updated the caption of Figure 3 to mention the distance ranges.

Figure 4.  If I draw the raypaths as full lines, then the figure becomes saturated and we learn little about the relative raypath densities.  Thus, I have plotted only every 21rst segment of each line using GMT dashed lines.  This makes the raypaths difficult to pick out but shows the coverage well.  I have changed the text and caption to be clearer about this.

Figures 5 and 6.  Southeast Iran does only have two stations, but it has many earthquakes.  This is why I get coverage there, especially for the longer Pn raypaths.  Figure 4 shows that the coverage is weaker in southeast Iran, but it is still sufficient.  Unfortunate that the coverage didn’t extend to the Makran coast.

Figure S2.  Yes, the event delays overprint each other because there are so very many events but I’m at a loss for how else to display this.  The figure does show the earthquake distributions.  The difficulty in getting much from this figure is why I put it in the supplement.

I now explain the checkerboard test in line 289.  The checkerboard test is common.  Checkerboard resolution tests are where we invert a synthetic data set, including noise, for a set of square anomalies alternating between low and high.  The smallest set of squares that can be imaged shows the minimum resolution length.  For all phases, squares of 2 degrees are well resolved. 

Figure S4.  Yes, again this picture is overcrowded because of the vast number of events and it is not so useful.  That’s why I put it in the supplement.  Often, authors just leave out the station and event delay pictures because they seem to only tell us where the stations and events occur and really aren’t very informative.  But I wanted to be complete in documenting my study.

Line 218 (now 291): Yes, it’s not appropriate to call this the Zagros so I crossed that out.

Line 220 (now 293):  Raypaths in the NW Alborz, from Figure 4, are dense for Lg amplitudes and the checkerboard test shows that 2 degree features can be resolved.  The high attenuation feature in the NW is about that size.  What worries me more is that it is on the edge of the tomography and this makes it harder to image so I crossed that section of the sentence out.

Line 224.  I was not aware of Kaviani et al 2020.  Thank you for this reference.  Their 15 km depth velocity section (their Figure 4) agrees with the phase velocity map I made for the Sg including the UDMA.  I don’t think their Figure 7 shows high velocity in the crust and they never specifically comment on it.  But more to the point, could the crustal root beneath the SSZ be causing crustal attenuation?  This is indeed a possibility that I now mention on line 534; however, this should cause lower group velocity and I don’t see that.

Line 225 (283):  Differences occur between estimates of spreading and Q and this study because the spreading and the Q values were adjusted to be those implied by the southern California magnitude formula 6.  This formula is also more in line with those of Table 1.  Thus, there is an inherent tradeoff between the absolute values of spreading and 1/Q.  I put this in the text when I discuss the data processing.  It is an important point.  However, spreading constants depend very much on the raypaths and are difficult to predict from theory.

Lines 276 (330): This is correct.  Most the Pg do not propagate across the southern Caspian Sea because they are too short.  The text has been corrected in those places.

Line 227:  I moved this to line 572 in the geologic discussion of the UDMA.

Depth and elevation were accounted for in the raytracing for the travel times.  I put this in line 160.  In addition, for raytracing there are always station and event terms.  For travel times, the event terms take up any error in depth and origin time.  For amplitudes, the event terms take up error in magnitude and station terms account for any station gains.  Yes, I’m sure there are many location errors in this data set, but the large volume of data overcomes that.

The reviewer says: “Please explain if it is possible to relate the body wave group velocities to the characteristics of inner-Earth features. As far as I know, it is not usual data set for tomography; so, please add a new section in the manuscript and explain how estimation of body wave group velocities can be useful for extracting information from the Earth .” Group velocity is new and a difficult one to explain so I introduced it in the introduction and then deferred its interpretation to the Geophysical Interpretation section where I devote two paragraphs to it.  I put a few extra sentences in those paragraphs to make it clear how to interpret group velocity in terms of earth structure.  The use of group velocity is to separate intrinsic and scattering attenuation.

Line 324: I referenced Bavali et al 2016.

Line 335: I don’t see a for in this line.  Although I am going by the original pdf I submitted I think the line index messed up.

Line 346:  The standard errors for the initial line fits of equation 1 are ridiculously small (on the order of velocity errors of 0.0002 km/s and intercept errors of .02 km/s).  A similar story applies to the initial estimation of Q from equation 3 (errors in Q on the order of 10 and errors in spreading on the order of .01).  This is because I have inverted tens of thousands of data points for only two or three parameters.  This does not mean that the rms spatial velocity variations or Q variations are small but only that their averages are extremely well determined.   These errors are rather meaningless parameters thus I put a note to this in line 286.  Perhaps more to the point is what are the errors in the final tomography figures?  I investigated this by putting the appropriate amount of noise into the checkerboard tests.  I’ve calculated the standard error of the velocity variations in past projects and it’s always small because of the damping.  In seismic tomography it’s the resolution, not the errors, that matters.

Line 371:  Again, the line numbers seem to be different between what I sent in and what the reviewer is looking at.  I believe he’s looking at lines 380-383 “Pn phase travel time velocities….”.  I put that sentence latter in the paragraph.

Line 371:  The reviewer suggests that 150 km is too low a limit for Pn distances.  Actually that was a mistake and it should have been 100.  It is true that Pn may not be a first arrival until after 150 km because of the crustal thickness; however, all these arrivals were specifically identified as Pn, so presumably the analyst could often identify Pn as the second arrival.  I put line 462 in to say this.

Line 387 (I find it on 400??).  Q sub kappa is the bulk modulus Q.  Another reviewer commented about the appropriateness of these equations so I just took them out.  I really don’t like the Poisson solid assumption anyway.  I did leave in the fact that P waves show higher attenuation since I think this is an important observation.

Line 394:  Yes, the minus sign should be a plus.

Line 414-419:  I added a couple of sentences to discuss the leakage of Lg into the mantle (line 534).  That sort of attenuation should come under scattering attenuation.  I didn’t put this in this paper, but in my study of the US I didn’t see much correlation between topography and attenuation.

Line 438:  Yes, I modified that sentence.  Sn phase data are questionable and I said that earlier on, but the group travel time plot makes it clear that a substantial number of Sn phases are seen and are often the maximum magnitude.

Line 461:  Yes, I removed the and I found on 474.

Line 450:  My lower Pn velocity beneath the UDMA is also found in Lu et al. 2012 in most as well as Amini et al 2012.  Amini et al 2012 also have low velocities in their teleseismic tomography. Motaghi et al. 2015 show a cross section with lower velocity beneath the SSZ and UDMA.  The reviewer mentions Paul et al 2010.  They do a combined receiver function / surface wave inversion.  They only have big blocks and show that the uppermost mantle goes from a high velocity block in the southwest to a low velocity block in the northwest beneath the UDMA in their southeastern Zagros cross line and is low velocity in their northwestern cross line (their Figure 6) – this is not too much out of line with what I see as the contrast between the Arabian Plate and the Iranian plate.  However, due to their block size I am resolving finer features.  Motaghi et al. 2017a do comment that the Arabian Plate underthrusts the SSZ and UDMA, but do not have the resolution I have.  Pn rays dive into the mantle and are not affected by most mantle roots.  Overall, I am confident of the lower Pn velocities beneath the UDMA.

Line 461: I removed the and.

Geodynamic models:  I’m not really sure which geodynamic models are being referred to here.  The dropping off of the Arabian slab really doesn’t affect the Pn.

Manuscript title:  I changed it to “2D Attenuation and velocity tomography of Iran”

Reviewer 3 Report

see attached pdf

Author Response

Reviewer 3:

This is a paper that uses raw seismic bulletin data to learn something new about the earth.  It is not a theoretical paper.  It uses the basic and simple concepts of attenuation and Q as they have been classically defined and widely accepted in seismology and physics for over 50 years (see Aki and Richards 1980; Sato, Fehler, and Maeda 2012).  The description of seismic attenuation as having absorptive and scattering components also goes back quite some time (see the book by Sato, Fehler, and Maeda, 2012).  However, the reviewer here is known for his dislike of the Q description of attenuation and has even published a book, “Seismological attenuation without Q”.  I appreciate his investigations into attenuation; however, his descriptions of attenuation are too sophisticated to be applied to crude seismic bulletin data.  I’ve tried to respond to the reviewers concerns here but I need to stick with the traditional viewpoints on Q to make sense of this data in a geologic context and to compare it to other studies.

I rearranged paragraph one and added some sentences in the fourth paragraph to more clearly define what I mean by spreading and attenuation.  From the traditional viewpoint, the difference between attenuation and 1/Q is just the pi/v/T coefficient of the exponent (T is period). 

Yes, the attachment of the model to the southern California model is unfortunate and biases the absolute attenuation and spreading values, but there was nothing else I could do with the data.  It’s the attenuation and velocity patterns that matter the most.

This tomography never measures path Q but starts from the raw amplitude.  But do the Q values on the maps represent what is normally thought of as material Q?  Are they comparable to Q values that would be measured in a hand sample?  These are the important questions the reviewer is asking.  I contend they do represent material Q (averaged by the resolution of the tomography!) as the patterns revealed match those of regional geology.  In this and other tomographies, I am impressed by how high attenuation is aligned with volcanic regions and basins.  Other tectonic divisions are also imaged.  In this paper it is also clear that the attenuation patterns are closely related to the velocity patterns.  If Q were just a random path phenomenon the tomographic attenuation images would be flat and no correlations between images would exist.  Furthermore, the group velocity maps also make geologic sense and match the patterns found on both the attenuation and phase velocity maps.  This too, would not occur if the group travel time were merely a path phenomenon.  I sincerely wish I had laboratory hand sample results on rock attenuation to compare to but there is no such compilation of data outside of the oil industry (I’ve participated in such experiments).  Even qualitatively comparative lab estimates between volcanic Q, sediment Q, and crystalline rock Q would be great.

I dropped line 77.  It really didn’t say what I wanted it to.  But, in disagreement with the reviewer, I interpret scattering Q to include large scale crustal variations including those in thickness, topography, and other features and I said this in my revision (line 109).  Seismic scattering Q is always discussed with respect to larger local or regional distance scales because that’s where the data is.  Takahashi’s papers on scattering that I reference view scattering as a crustal scale phenomena for local phases and model it as such.  Some of the other papers I reference use Takahashi’s results with data to infer crustal scale scattering under Japan and elsewhere.

The reviewer is right the teleseismic Pn at distances over 3000 km is what is being discussed by Nielsen and Thybo and does not apply here.  I removed this reference from the paper.

I got rid of the section on bulk and shear attenuation from Anderson’s book.  I don’t think the assumption of a Poisson media applies anyway and the section is not needed.

Usually, the terms phase velocity and group velocity are applied to only to surface waves because the distinction hasn’t been needed for body waves.  I’ve had to extend these definitions to body waves for lack of better terminology.  The phase velocity is that given by the wave equation (eg sqrt(mu/rho) or sqrt((lamda+2Mu)/rho)) when applied to body waves (see Richter 1958).  That is exactly what is being measured by the phase travel times.  The term phase velocity is often used in seismic velocity tomography papers.  The group velocity is the speed which the amplitude envelope moves, and that again is exactly what is being measured.  I’ve stuck with these terms because I think they are clear to most seismologists and I can’t think of better terms.  I made changes in line 88 to make these definitions clear.  Both phase and group velocity depend on the bandwidth, of course, but bandwidth is fixed here.

tg = sgx + ag. The reviewer points out here that ag will be a function of x (it is the intercept and depends on depth and elevation as well as velocities).  This is true, but in general these terms do not vary much geographically.  In the tomography, the geographical variation of ag will be taken care of by the station and event correction terms.

The delay time is explained by primarily by scattering and this is not wrong.  The delay between the peak-amplitude and onset times was extremely well studied and modeled by Takahashi in a series of cited publications as well as being examined in the Sato and Fehler 2012 book.  I also cite with seven other publications that apply this interpretation of scattering to real Earth data on crustal scales.  The conclusion is that this “peak delay time” (Takahashi’s term) depends on scattering and is relatively insensitive to intrinsic attenuation.   The source signature is simply a static shift in the peak delay time and, yes, without scattering we would have much sharper wave onsets.  As a counter example, lunar peak delay times are very long because of the extreme scattering.  Finally, I note that the imaged group velocity images correlate well with geology and the other images thus indicating the group travel times are not random.  Interpreting the group velocities in terms of phase velocity and scattering is well supported by the literature as well as making geologic sense.

I’ve checked the equations to get the numbering, italics, and periods in the correct places.

Round 2

Reviewer 2 Report

Thank you for including the comments.

Reviewer 3 Report

I only suggest correcting a misspelled 'iteratively winnowed' in line 203.

We can of course debate forever whether the "Q" recalculated from amplitudes and produced by scattering and refractions on regional-scale structures is actually related to the material of the volcanics or just to topography or crustal thickness.